# An Improved Ambiguity Resolution Algorithm for Smartphone RTK Positioning

**DOI:** 10.3390/s23115292

**Published:** 2023-06-02

**Authors:** Yang Jiang, Yuting Gao, Wei Ding, Fei Liu, Yang Gao

**Affiliations:** 1Department of Geomatics Engineering, University of Calgary, Calgary, AB T2N 1N4, Canada; ygao@ucalgary.ca; 2College of Geomatics, Xi’an University of Science and Technology, Xi’an 710054, China; ygao@xust.edu.cn; 3School of Geomatics, Liaoning Technical University, Fuxin 123000, China; dingwei@lntu.edu.cn; 4Profound Positioning Inc., Calgary, AB T2P 3G3, Canada; fliu@profoundpositioning.com

**Keywords:** smartphone positioning, real-time kinematic (RTK), ambiguity resolution (AR), global navigation satellite system (GNSS)

## Abstract

Ambiguity resolution based on smartphone GNSS measurements can enable various potential applications that currently remain difficult due to ambiguity biases, especially under kinematic conditions. This study proposes an improved ambiguity resolution algorithm, which uses the search-and-shrink procedure coupled with the methods of the multi-epoch double-differenced residual test and the ambiguity majority tests for candidate vectors and ambiguities. By performing a static experiment with Xiaomi Mi 8, the AR efficiency of the proposed method is evaluated. Furthermore, a kinematic test with Google Pixel 5 verifies the effectiveness of the proposed method with improved positioning performance. In conclusion, centimeter-level smartphone positioning accuracy is achieved in both experiments, which is greatly improved compared with the float and traditional AR solutions.

## 1. Introduction

High-precision smartphone positioning is increasingly demanded to enable potential applications such as lane-level vehicle navigation, augmented reality walking/driving, and precise agriculture via phones [1,2,3]. Although real-time kinematic (RTK) techniques with ambiguity resolution (AR) have been widely used for high-end Global Navigation Satellite System (GNSS) receivers, advanced positioning algorithms are required to deal with a much-poorer-quality pseudorange and carrier phase measurements from smartphones [4,5,6]. First, the high and unstable stochastic properties of the measurements, as well as their possible outliers, pose challenges since they often affect the precision of float RTK solutions and reduce the ambiguity-fix success rate [7]. Second, smartphone initial phase bias (IPB) will corrupt the integer property of the carrier ambiguities [8,9,10]. Third, the biases caused by smartphone antennas and carrier phase multipath effects would also affect the position determination [11,12]. As a result, some smartphone carrier phase ambiguities are contaminated with ambiguity biases, making fixed solutions significantly biased [13].

Many studies have been conducted to explore AR with smartphone GNSS measurements. After assessing the observation quality of smartphone pseudorange and carrier phase measurements, Paziewski et al. [14] found that a stochastic model could be developed based on signal-to-noise ratio measurements and obtained results better than the traditional elevation-dependent model. Gao et al. [15] proposed using the raw measurement’s standard deviation and the multipath indicator provided by the Android application programming interface to improve the performance of smartphone AR. In addition, many studies have been undertaken to deal with these issues in realistic smartphone applications, such as multipath [16,17] and antenna offset problems [18,19], and their impacts on carrier phase ambiguities are widely noticed. To resolve the IPBs, Geng et al. [9] proposed a method for post-processing calibration. Yong et al. [20] concluded that the antenna offsets could be minimized by keeping the smartphone in an upright position. Additionally, the implementation of partial AR (PAR) can reduce the impact of the ambiguity biases on the fixed solutions [21,22,23], as confirmed by [24,25]. Although many studies have been undertaken to assess the performance of smartphone AR, few implementations focus on kinematic experiments in real-time smartphone applications. However, with the user motion, the time-varying multipath effects and antenna offsets increase the difficulty of handling the carrier phase ambiguity biases, which in turn pose challenges for smartphone kinematic AR. The precision of float solutions is also lower in kinematic applications due to the unstable stochastic properties of measurements, which generally decrease the ambiguity–fix success rate.

This study focuses on improving the AR algorithm for smartphones to achieve high-precision smartphone GNSS positioning. Specifically, the integer ambiguities are resolved by coupling the search-and-shrink procedure with testing methods, including the multi-epoch double-differenced (DD) model residual test and the ambiguity majority tests for candidate ambiguities and vectors. To verify the proposed method, two smartphone experiments are conducted, where Xiaomi Mi 8 is used for a static test, and Google Pixel 5 is used for a kinematic test.

Section 2 presents the methodology used in this study. Section 3 introduces the static dataset and its evaluation outcomes of AR efficiency. Section 4 discusses the kinematic positioning performance. The conclusions are summarized in Section 5.

## 2. Methodology

To realize AR for kinematic applications with single- or dual-frequency GNSS, including smartphones, the GNSS community uses the popular method of LAMBDA [20,26]. A unimodal transformation and a search-and-shrink scheme based on the integer least-squares (ILS) principle find the optimal ambiguity integers with a real-time computational load [27]. Until recently, the method of best integer equivariance (BIE) has proven to be a better replacement [28], where the AR performance is optimized in the sense of minimizing the mean square error [29,30,31]. However, these methods, including BIE, naturally consider the input float ambiguity estimations to be unbiased integers. This does not fit in the application of smartphone AR since the existence of IPB, carrier phase multipath effects, and antenna offsets lead to non-negligible ambiguity biases. As a result, the unimodal transformation and the search-and-shrink scheme are inaccurate and likely to produce a set of incorrect ambiguity integers [31]. Although PAR coupled with improved ambiguity validation strategies, such as protection-level, are proposed [32,33], they are generally not sufficiently efficient to identify the correct ambiguity integer set for smartphones due to the volume of such ambiguity biases and the significant measurement noises.

Primarily, in this study, the AR scheme using the search-and-shrink procedure is applied without the unimodal transformation, where only one integer ambiguity is resolved at a time. To select the optimal ambiguity candidate, the methods of ambiguity majority tests, as well as the ambiguity validation with the multi-epoch DD residual test are applied based on the candidate ambiguity vectors by LAMBDA with the ILS principle [34]. In the following subsections, the commonly used GNSS mixed-integer model, the AR scheme of search-and-shrink, the ambiguity majority tests, and ambiguity validation with the multi-epoch DD residual test is explained in detail.

### 2.1. GNSS Mixed-Integer Model

For smartphone RTK based on the DD model with an acceptable baseline length, most GNSS error sources, such as the ionosphere, troposphere, and others, are eliminated. Therefore, the linearized single-epoch observation equation is a mixed-integer model, including parameters of integer-valued carrier phase ambiguities and the real-valued baseline vector [35]:(1)y=Aa+Bb+e,
where $y\in R^{m}$ refers to the measurement vector contaminated by a zero-mean normal-distributed noise $e\sim N(0,Q_{yy})$; $a\in Z^{n}$ and $b\in R^{p}$ denote the integer-valued ambiguities and the real-valued baseline vector, respectively, and $A\in R^{m\times n}$ and $B\in R^{m\times p}$ represent their respective design matrices. With least-squares or Kalman filtering, the float solution discards the integer nature of ambiguities [35]:(2)$$\left[\begin{array}{c} \hat{a}\\ \hat{b} \end{array}\right]\sim N\left(\left[\begin{array}{c} a\\ b \end{array}\right],\left[\begin{array}{cc} Q_{\hat{a}\hat{a}} & Q_{\hat{a}\hat{b}}\\ Q_{\hat{b}\hat{a}} & Q_{\hat{b}\hat{b}} \end{array}\right]\right),$$
where $\hat{a}$ and $\hat{b}$ denote the float solutions with respect to ambiguities and coordinates, and $Q_{\hat{a}\hat{a}}$, $Q_{\hat{a}\hat{b}}$, $Q_{\hat{b}\hat{a}}$, and $Q_{\hat{b}\hat{b}}$ represent the variance and covariance components of the estimated parameters. To solve for the integer ambiguities $\overset{˘}{a}=I(\hat{a})$, many integer equivariance estimators can be used, such as ILS and BIE [28]. In this case, ILS is discussed, which provides ${\overset{˘}{a}}_{ILS}$:(3)$${\overset{˘}{a}}_{ILS}=argmin{\left\|\hat{a}-z\right\|}_{Q_{\hat{a}\hat{a}}}^{2},\forall z\in Z^{n};$$

If the integer ambiguities are accepted by ambiguity validation methods, such as a ratio test, a fixed solution can be achieved by readjusting $\hat{b}$, which gives $\overset{˘}{b}$:(4)$$\overset{˘}{b}=\hat{b}-Q_{\hat{b}\hat{a}}Q_{\hat{a}\hat{a}}^{-1}\left(\hat{a}-\overset{˘}{a}\right);$$

Although for smartphones, ambiguities suffer from biases, principles such as ILS are valid for describing the integer components; therefore, they are mostly unbiased. In this way, those biases are absorbed by the carrier phase measurement noises, and the magnitude of the residuals contributes to larger position errors, e.g., 3 cm to 10 cm. Overall, the fixed solutions should remain bias-free and with high precision. However, to solve the ILS principle of (3), the popular search-and-shrink procedure is widely used to reduce the computational complexity of the traditional methods [36,37], which can cause ambiguous candidate vectors to be biased on smartphones. The detailed algorithm and the reasons are explained in the following subsection.

### 2.2. Search-and-Shrink Procedure for Ambiguity Resolution

For a two-dimensional (2D) example where the ambiguities are $a_{1}$ and $a_{2}$, the search-and-shrink procedure calculates the conditional ambiguity of $a_{1}$ once if ${\hat{a}}_{2}$ is rounded to an integer ${\hat{z}}_{2}$ by bootstrapping using [25]:(5)$${\hat{a}}_{1|2}={\hat{a}}_{1}-\sigma_{{\hat{a}}_{1}{\hat{a}}_{2}}\sigma_{{\hat{a}}_{2}{\hat{a}}_{2}}^{-1}\left({\hat{z}}_{2}-{\hat{a}}_{2}\right),$$
where ${\hat{a}}_{1|2}$ denotes the conditional ambiguity of $a_{1}$, $\sigma_{{\hat{a}}_{1}{\hat{a}}_{2}}$ represents the covariance of $a_{1}$ and $a_{2}$, and $\sigma_{{\hat{a}}_{2}{\hat{a}}_{2}}$ signifies the variance of $a_{2}$. Then, with the conditional ambiguities, the boundary value of ${\left\|\hat{a}-z\right\|}_{Q_{\hat{a}\hat{a}}}^{2}$ is reduced so that a smaller search space can be reached. Iteratively, the search space shrinks to reach the k integer ambiguity candidate vector, where k is user specified. In the sequel, $\overset{˘}{A}$ is used to denote the combination of candidate vectors:(6)$$\overset{˘}{A}=\left[\begin{array}{ccc} {\overset{˘}{a}}^{1} & {\overset{˘}{a}}^{2} & \begin{array}{cc} \cdots & {\overset{˘}{a}}^{k} \end{array} \end{array}\right],\;\overset{˘}{A}\in Z^{n\times k},$$

Commonly, many ILS-based algorithms have applied this procedure [35,38], where the final decision is made as ${\overset{˘}{a}}^{1}$ after the ambiguity validation. For the traditional implementation of the LAMBDA method, vector $\hat{a}$ is sorted by the increasing order of the variances of its elements, meaning $\sigma_{a_{1}a_{1}}\leq \sigma_{a_{2}a_{2}}$, where the initial search space is defined as follows:(7)$$F\left(z\right)={\left\|\hat{a}-z\right\|}_{Q_{\hat{a}\hat{a}}}^{2}\leq \chi^{2},$$
where z, with the minimum value of function $F\left(z\right)$, is the optimal ILS solution $\overset{˘}{a}$, and $\chi^{2}$ can be predetermined and can also shrink during the search [39,40]. LAMBDA is coupled with a unimodular transformation based on the $L^{T}DL$ factorization of $Q_{\hat{a}\hat{a}}$ so that the search process can be more efficient [39].

However, in the case of smartphone carrier phase measurements, many ambiguities are biased and have lost their integer nature. Therefore, the above-mentioned search-and-shrink procedure is less valid because it assumes an integer grid. Specifically, if $a_{2}$ is naturally not an integer, the conditional ambiguity of ${\hat{a}}_{1|2}$ can be affected by the ambiguity bias of $a_{2}$. For higher-dimension problems, those ambiguity biases can accumulate while the search space shrinks, causing the further search space to be incorrect. Furthermore, coupling with the unimodular transformation can expose such bias to even more original ambiguities after the back transformation. In this case, current AR methods with the search-and-shrink procedure, such as LAMBDA, frequently produce an incorrect set of integer ambiguities. Unfortunately, an improved procedure considering such ambiguity biases is currently unavailable. This study primarily focuses on selecting the ambiguities to be conditionally rounded, without unimodular transformation, to mitigate the impact of ambiguity biases.

### 2.3. Ambiguity Majority Tests for Candidate Ambiguities and Vectors

Although LAMBDA with unimodular transformation can spread the impact of ambiguity biases, it is expected that some ambiguities would not vary much among different candidate vectors. This is because their ambiguity variances are relatively small due to better observation conditions; therefore, they are sorted in front of the ambiguity vector, which also makes them less biased by the search-and-shrink procedure. Therefore, these ambiguities are more trustworthy to be integers. Then, we innovatively calculate the modulus of each ambiguity $a_{i}$, namely $M_{a_{i}}$, from the combined candidate vector $\overset{˘}{A}$ by the LAMBDA method, these were formulated as follows:(8)$$M_{a_{i}}=mod{\left({\overset{˘}{A}}_{i,j}\right)}_{j\in \{1,\ldots ,k\}},$$
where mod(·) denotes the modulo operation. Therefore, the number of candidates of ambiguity $a_{i}$ that equals $M_{a_{i}}$, namely $N_{a_{i}}$, can be calculated as follows:(9)$$N_{a_{i}}=\sum_{j\in \{1,\ldots ,k\}}\left({\overset{˘}{A}}_{i,j}=M_{a_{i}}\right);$$

The value of $N_{a_{i}}$ reflects the confidence in the selected $M_{a_{i}}$. Straightforwardly, with a higher value of $N_{a_{i}}$, the corresponding ambiguity can be more trusted since it is less variant among different ambiguity vectors. As a drawback, with a k value larger than 2, some candidate vectors by the LAMBDA method can be too abnormal to introduce them to the majority test, which can reduce its accuracy. Here, the value of ${\left\|\hat{a}-z\right\|}_{Q_{\hat{a}\hat{a}}}^{2}$ is not used to weigh the candidates for the majority test since it is potentially affected by the ambiguity biases and cannot be accurate. To exclude the abnormal candidate ambiguities, we apply the novel majority test to each candidate ambiguity vector, which was formulated as follows:(10)$$N_{{\overset{˘}{a}}^{j}}=\Sigma_{i=1}^{n}\Sigma_{l=1;l\neq j}^{k}\left({\overset{˘}{A}}_{i,l}={\overset{˘}{A}}_{i,j}\right),$$
where $N_{{\overset{˘}{a}}^{j}}$ denotes the majority test for a candidate ambiguity vector ${\overset{˘}{a}}^{j}$. With the larger value of $N_{{\overset{˘}{a}}^{j}}$, ${\overset{˘}{a}}^{j}$ can be expected to be closer to the center of the candidates; therefore, it is a reasonable metric to roughly measure its correctness. It is also noted that this algorithm shown in (8)–(10) performs a partial integer bootstrapping method [20,41]. To further increase its accuracy, the method of multi-epoch residual test is used for ambiguity validation, which excludes those abnormal candidate vectors with large loss values before applying this ambiguity majority test, as shown in the next subsection; After that, (9) can be applied to finally determine one ambiguity to be fixed. After updating the float ambiguities using (5), a new round of majority test can trigger. Iteratively, the full ambiguity set is resolved into integers, and the fixed solutions are obtained.

### 2.4. Multi-Epoch Residual Test for Ambiguity Validation

For multiple adjacent GNSS epochs of (1), the integer part of a will be identical as long as no cycle slips occur. For kinematic applications of the smartphone, the non-integer part of b is time-dependent. Therefore, we write a multi-epoch combined DD observation model as follows:(11)$$y_{t_{1,w}}=Aa+B_{t_{1},w}b_{t_{1},w}+e_{t_{1,w}},Q_{y_{t_{1,w}}},$$
(12)$$y_{t_{1,w}}=\left[\begin{array}{c} \begin{array}{c} y_{t_{1}}\\ y_{t_{2}} \end{array}\\ \begin{array}{c} \vdots \\ y_{t_{w}} \end{array} \end{array}\right],\;B_{t_{1},w}=\left[\begin{array}{cc} \begin{array}{cc} B_{t_{1}} & \\ & B_{t_{2}} \end{array} & \\ & \begin{array}{cc} \ddots & \\ & B_{t_{w}} \end{array} \end{array}\right],\;b_{t_{1},w}=\left[\begin{array}{c} \begin{array}{c} b_{t_{1}}\\ b_{t_{2}} \end{array}\\ \begin{array}{c} \vdots \\ b_{t_{w}} \end{array} \end{array}\right],\;e_{t_{1,w}}=\left[\begin{array}{c} \begin{array}{c} e_{t_{1}}\\ e_{t_{2}} \end{array}\\ \begin{array}{c} \vdots \\ e_{t_{w}} \end{array} \end{array}\right],$$
where the epoch times $t_{1}$, $t_{2}$, ⋯, $t_{w}$ are adjacent and w defines the window size of the model. Here, with the total number of measurements m×w, the number of estimation parameters are n ambiguities and p×w user coordinates. Hence, with a larger value of w, the estimation redundancy grows, and the model complexity also increases. For ambiguity validation, the integer part of a is provided from a candidate vector ${\overset{˘}{a}}^{i}$, where the non-integer part can be estimated using least squares, denoted as ${\hat{b}}_{t_{1,w}}^{i}$, as well as the measurement residuals, denoted as ${\hat{r}}_{t_{1,w}}^{i}$:(13)$${\hat{b}}_{t_{1,w}}^{i}={\left(B_{t_{1},w}^{T}Q_{y_{t_{1,w}}}^{-1}B_{t_{1},w}\right)}^{-1}B_{t_{1},w}^{T}Q_{yy_{t_{1,w}}}^{-1}\left(y_{t_{1,w}}-A{\overset{˘}{a}}^{i}\right),$$
(14)$${\hat{r}}_{t_{1,w}}^{i}=y_{t_{1,w}}-A{\overset{˘}{a}}^{i}-B_{t_{1},w}{\hat{b}}_{t_{1,w}}^{i},$$

Therefore, the problem of ambiguity validation can be interpreted as an overall residual test based on ${\hat{r}}_{t_{1,w}}^{i}$, which can be calculated as follows [42,43,44]:(15)$$T_{q}=\frac{{\left\|{\hat{r}}_{t_{1,w}}^{i}\right\|}_{Q_{y_{t_{1,w}}}}^{2}}{q};$$
where $T_{q}$ denotes the overall test statistics, and q represents the degree of freedom, in this case, $q=\left(m-p\right)\times w$. In the sequel, the calculated $T_{q}$ is used as the residual test statistics of the DD model, called DD residual test statistics. It should be noted that considering the existence of cycle slips, a practical implementation of this method should adaptively reduce the window size for satellites subjected to the cycle slip detection. In addition, to determine a proper window size w, the time correlation of the carrier phase measurements should be considered depending on the GNSS device [45,46]. In the case of the experiment smartphones, Google Pixel 5 and Xiaomi Mi 8, this study used w=10 s, to balance the algorithm efficiency and real-time computational load.

Detailed algorithm implementation parameters are summarized in Table 1. For each iteration, 10 candidate ambiguity vectors are generated using the LAMBDA method, which is reduced to 4 candidates by both the majority test using (10) and residual test using (15). Then, the single fixable ambiguity can be determined using (9).

In the following subsections, a static experiment is presented to show the efficiency of the proposed method in smartphone AR, followed by a kinematic experiment to demonstrate its positioning accuracy.

## 3. Static Experiment—Smartphone Ambiguity Resolution Efficiency

To evaluate the performance of the proposed method in terms of AR, this subsection focuses on an experiment based on the Xiaomi Mi 8 smartphone. Although previous studies widely use external GNSS antennas and signal repeaters to enhance smartphone observation quality, this study is based on the smartphone as it is, where no external antennas and signal repeaters are applied. To extract the evaluation details such as ambiguity biases, this experiment is static, with the smartphone in the upright position and with known reference coordinates. First, the data show the number of satellites and a skyplot. Second, the positioning solutions are discussed, where the proposed AR method is compared with the LAMBDA method with a full ambiguity resolution (FAR) strategy [47], r-ratio ambiguity validation with a 2.0 ratio threshold [48] due to the generally lower success rate on smartphones, and the float solutions. Later, the efficiencies of the ambiguity majority test and the DD residual test are demonstrated. Last, with the fixed smartphone position and the resolved ambiguity integers, the estimated ambiguity biases are captured and analyzed. In the sequel, the error statistics of root-mean-square (RMS), standard deviation (STD), and mean values are commonly used.

This experiment was conducted in Calgary, Canada, at the local time of 1 PM, 5 March 2021, where the software Geo++ RINEX Logger, version 2.4.3, is used to collect the 1-Hz GNSS data from Xiaomi Mi 8. The base station receiver is a Trimble NetR9, with a baseline length of 9.23 km, which logs at the rate of 1 Hz. Overall, the dataset includes 5261 GNSS epochs, with a total duration of 87.7 min. For data processing, two GNSS constellations, GPS and Galileo, are used, with the signals of GPS L1 C/A and L5 (Q) and that of Galileo E1 (C) and E5a (Q), with an elevation cut-off angle of 4° and a signal-to-noise-ratio threshold of 10 dB-Hz. For the float ambiguity solutions, a Kalman filtering scheme with kinematic parameter settings is used, shown in Table 2, which can be found in Takasu and Yasuda [49]. Additionally, the ionosphere and troposphere model corrections, referring to the Klobuchar model and the Saastamoinen model with the Neill mapping function, respectively, are applied to the measurements of (1) in advance [50,51,52]. After the double-differencing and these model corrections, ionosphere and troposphere errors can be considered to be eliminated in our processing [53].

For AR, the proposed method is used and compared with the LAMBDA method. Here, to avoid the antenna offset problem among signal frequencies, only the signals on the first frequency, that is, GPS L1 C/A and Galileo E1 (C), are used for AR.

Figure 1 reveals that, on average, Xiaomi Mi 8 has 10.1, 1.6, 5.5, and 5.2 satellites on the signals of GPS L1 C/A and L5 (Q) and that of Galileo E1 (C) and E5a (Q), respectively. Overall, the observed satellites are 13 to 18 in total, while GPS and Galileo have 8 to 11 and 5 to 7, respectively, which are sufficient for the AR experiment. As shown in Figure 2, G14, G28, G30, and E21 are mostly under high-elevation conditions.

Figure 3 provides the positioning error time-series of Xiaomi Mi 8 using the proposed method, compared with the LAMBDA method and float solutions. Although it is a static experiment, the data are processed based on kinematic filtering settings; therefore, the results are representative for kinematic applications as well. The static coordinates of the smartphone are calculated by static post-processing with AR, which gives the millimeter-level STD so it can be adopted as a reference. Generally, the proposed method reaches centimeter-level RMS values, which means 100% correct fixation. For comparison purposes, the LAMBDA method is significantly biased because it lacks consideration of the ambiguity biases and often provides incorrect ambiguity vectors. After a period of convergence, it reaches the correct ambiguity set, which took 3000 epochs. As a result, the LAMBDA method reaches an accuracy of 0.6 m, 0.7 m, and 1.3 m, which are worse than that of the float solutions, reaching 0.2 m, 0.2 m, and 0.1 m, respectively. As a primary conclusion, the proposed method provides the correct ambiguity set in most cases.

In Figure 4, the DD residual test is evaluated in terms of its efficiency. Here, the resolved ambiguity vector for each epoch is obtained by the proposed AR method, which is compared with the other candidate ambiguity vectors from the LAMBDA method in the majority test in terms of the DD residual test statistics. As shown, the test statistics of the candidates are mostly from 0.2 cm to 3.5 cm, while the resolved ambiguity vectors have DD residual test statistics that are mostly from 0.1 cm to 0.5 cm. For 74% of all epochs, the resolved ambiguity vectors have the lowest residual test statistics, which have the second lowest residual test statistics for 2% of epochs. In summary, the DD residual test is a promising way to select trustworthy candidate ambiguity vectors.

For the ambiguity majority tests, Figure 5 illustrates the AR efficiency by comparing the test values from the resolved ambiguity vector and the candidate ambiguity vectors from the LAMBDA method by the majority test, reflected by (9). At a glance, the resolved ambiguity vector gives higher test values than the other candidates in most cases, that is, for 3714 of all epochs (71%). For all epochs, the resolved ambiguity vector mostly ranks first to second among all candidate vectors, which is 1.3 on average. Therefore, it is reasonable to exclude all candidates whose majority test values are lower than the third.

After achieving AR, the ambiguity biases can be captured using the static reference coordinate of the smartphone and resolved ambiguities, as input by (1). Figure 6 provides the time-series of the captured ambiguity biases. Although satellite ambiguities with healthy conditions are witnessed, such as E15 and E19, it is evident that ambiguity biases exist, which can evolve over time, especially for G13, G17, G21, and E01. Moreover, it is observed that GPS satellites suffer half-cycle slips, where sudden jumps of 0.5 cycles can frequently occur; see G07 and G14 as examples. Similar performances can be seen in [9], which agrees with our converged LAMBDA solutions shown in Figure 3 from epoch 4500. However, these contribute to the ambiguity biases, making it less possible to correctly achieve AR in real-time applications. Consequently, it is reasonable that conventional AR methods that have excellent performances on common GNSS receivers, such as LAMBDA, have limited efficiency on smartphone applications because there are frequent ambiguity biases.

For a detailed explanation, Figure 7 presents the statistics of ambiguity biases for each satellite. Their RMS values reach 0.07 to 0.31 cycles, which are normally 0.03 to 0.05 cycles for GNSS receivers such as u-blox modules. For the mean values, they vary from 0.02 to 0.20 cycles, which means almost 0 cycles compared with u-blox. In other words, the existence of smartphone ambiguity biases cannot be ignored before achieving AR using current methods.

In summary, this static experiment with Xiaomi Mi 8 proves that the proposed method demonstrates a significant improvement in terms of AR efficiency. The results show that the search-and-shrink procedure coupled with the majority test and the DD residual test is efficient in obtaining the correct ambiguity vectors from candidates. It also proves the existence of ambiguity biases in smartphone GNSS data, which further demonstrates the necessity of the proposed method.

## 4. Kinematic Experiment—Smartphone Positioning Performance

The kinematic experiment uses the smartphone Google Pixel 5, whose ground trajectory and data collection platform are shown in Figure 8. Similar to the static experiment, we apply no external antennas or signal repeaters to perform data collection. As can be seen, an open area is selected, where dynamic ground multipath, antenna offsets and smartphone orientation variations are the primary causes of ambiguity biases. To provide positioning reference, two survey-grade GNSS antennas with u-blox ZED-F9P receivers are used to provide RTK fixed solutions with centimeter-level accuracy. In this way, the relative location of Google Pixel 5 can be described through two directions, that is, the along-track and the cross-track directions with respect to the antennas of GNSS receivers, which are denoted as Receiver #1 and #2, respectively. With a long-term calibration of the along-track and the cross-track offsets of Google Pixel 5, 36.51 cm and 9.90 cm can be obtained for their ground truth values, respectively. This has made us capable of conducting kinematic accuracy evaluations with high levels of confidence.

This experiment was conducted in Calgary, Canada, at the local time of 8 PM, 4 April 2022, where smartphone GNSS data are collected by Geo++ RINEX Logger, version 2.4.3. The base station is equipped with a Trimble NetR9, with a 5 km distance from the smartphone. The dataset includes a total of 1771 s. The GPS, Galileo, and GLONASS constellations are used, where GLONASS satellites are unavailable for AR but are used as an additional source for geometry-based cycle-slip detection [55]. In this experiment, GLONASS is needed because, in the kinematic applications, cycle-slips are more frequent than the previous static applications, which this study should consider and minimize by increasing accessible satellites. During the kinematic experiment, the first and last 200 epochs are static when verifying the performance of Google Pixel 5 compared with Xiaomi Mi 8. The rest of the configurations are the same as the static experiment. In Figure 9, the numbers of satellites involved and the frequency signals are plotted, which are, on average, 9.8, 2.7, 9.8, 8.4, and 7.7 for GPS L1, GPS L5, Galileo E1, Galileo E5a, and GLONASS L1, respectively.

Figure 10 provides the 2D, along-track, and cross-track positioning errors during the experiment. In general, the solutions using the LAMBDA method are scattered, where the average error distance is 11.74 cm. For the proposed method, it is 4.6 cm, which indicates a significant improvement. For 95% of the data, the proposed method has an error distance within 10.2 cm versus 39.9 cm for the LAMBDA method. Therefore, it is concluded that the proposed method improves positioning accuracy by considering ambiguity biases in smartphones.

Figure 11 presents the time-series of the along-track and cross-track positioning errors, comparing the proposed method with LAMBDA and the float solutions. The RMS values of the float solutions are 9.8 cm and 8.1 cm for the along-track and the cross-track directions, in contrast to 12.8 cm and 11.7 cm for LAMBDA, respectively. Generally, LAMBDA shows lower performance than the float solutions due to the existence of ambiguity biases that frequently affect its search-and-shrink procedure; therefore, the resolved ambiguities are less trustworthy. For the proposed method, the optimized solution reaches accuracy values of 3.8 cm and 3.9 cm. In addition, it can be seen that the first and last 200 epochs have smoother solutions, and this is because, when static, the quality of smartphone GNSS measurements, including the noise levels of the carrier phases and pseudoranges [56], and satellite availability are relatively better [57]. Overall, it is evident that the proposed method outperforms the LAMBDA method and the float solutions in terms of AR and positioning accuracy.

## 5. Conclusions

This study proposes an improved AR algorithm for smartphone positioning by considering ambiguity biases, where the search-and-shrink method is used with the testing methods, including a multi-epoch DD model residual test and majority tests for candidate ambiguities and vectors. The static dataset is first applied to evaluate smartphone AR efficiency. Secondly, the kinematic data verify the improvement in smartphone positioning performance. The key points are summarized as follows:The existence of ambiguity biases is not negligible for AR based on smartphone devices. In the static experiment performed with Xiaomi Mi 8, the average level of ambiguity biases ranges from 0.07 to 0.31 cycles.The proposed AR scheme using the search-and-shrink procedure coupled with the majority test and the multi-epoch DD residual test can overcome the problem of AR. The majority test can identify the actual ambiguity vector from the candidates with an accuracy of 71% for the first rank and 6% for the second rank versus 74% and 2% for the DD residual test.The proposed method achieves AR to improve the positioning accuracy of smartphones. For the static test, the RMS values are 1.1 cm, 1.7 cm, and 2.1 cm for east, north, and upward directions, in contrast to 0.2 m, 0.2 m, and 0.1 m for the float solutions, respectively. For the kinematic test, the RMS values are 3.8 cm and 3.9 cm for the along-track and the cross-track directions versus 9.8 cm and 8.1 cm for the float solutions.

## Figures and Tables

**Figure 1 sensors-23-05292-f001:**
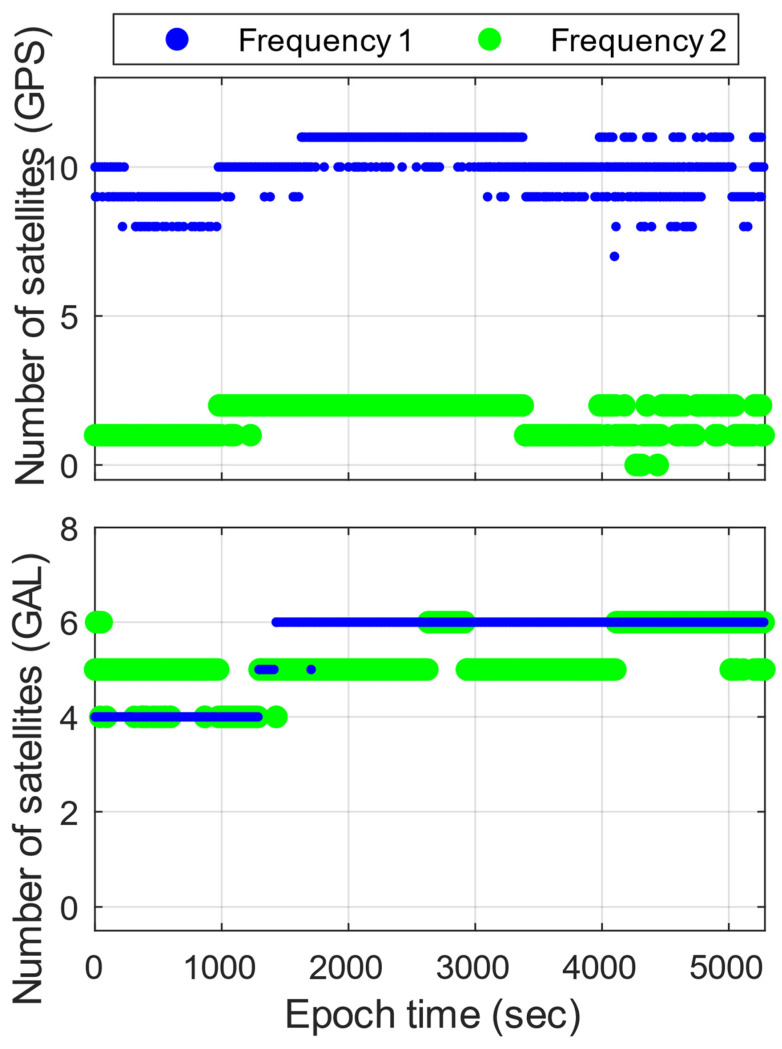
Numbers of GPS and Galileo (GAL) satellites of Xiaomi Mi 8 for dual frequency. For GPS, Frequency 1 is L1 C/A, and Frequency 2 is L5 (Q). For Galileo, Frequency 1 is E1 (C), and Frequency 2 is E5a (Q).

**Figure 2 sensors-23-05292-f002:**
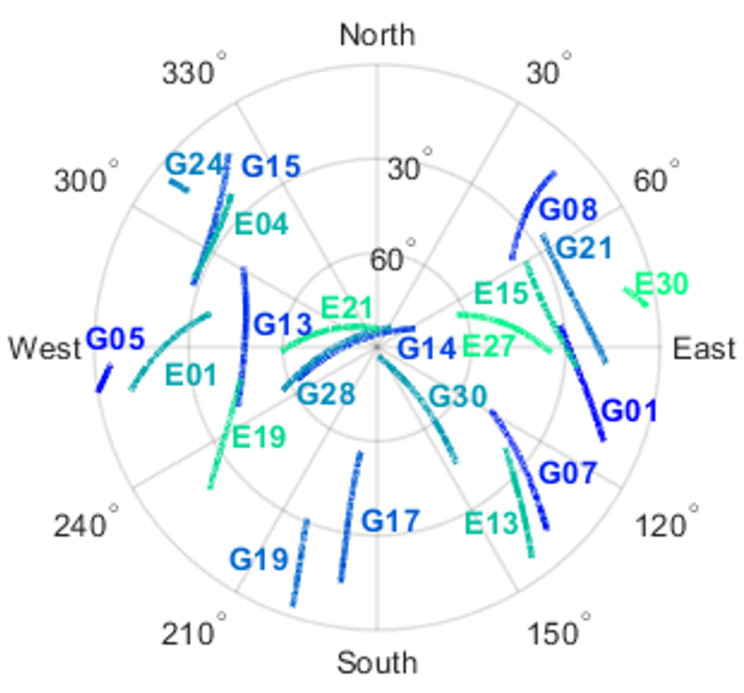
Skyplot of GPS and Galileo satellites of Xiaomi Mi 8.

**Figure 3 sensors-23-05292-f003:**
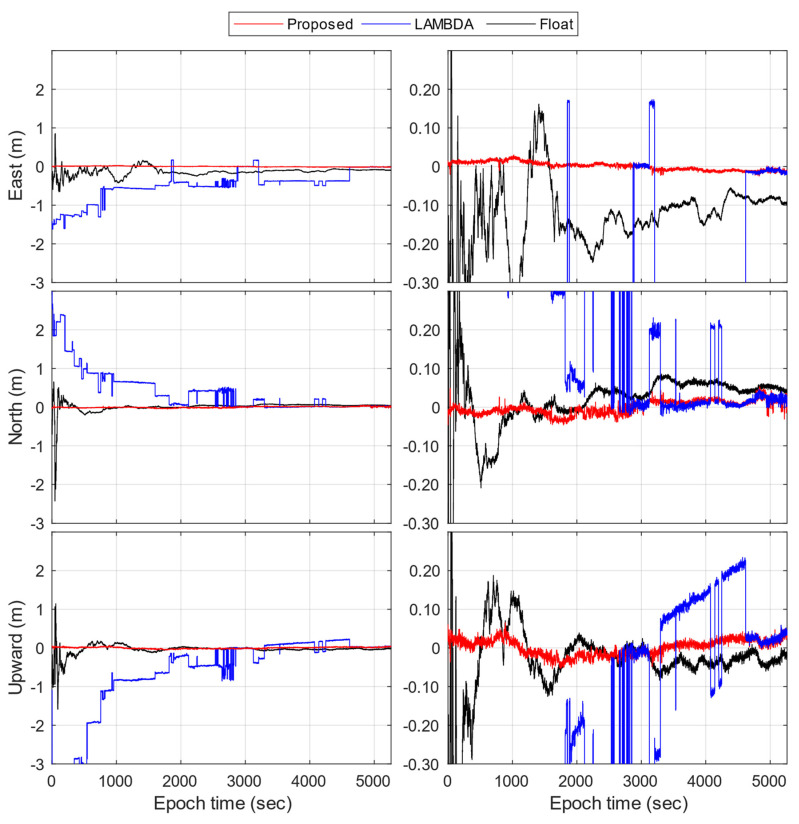
Positioning error time-series of Xiaomi Mi 8 by the proposed method, the LAMBDA method, and float solutions. The figures on the left are zoomed-out plots, while the figures on the right are zoomed-in plots.

**Figure 4 sensors-23-05292-f004:**
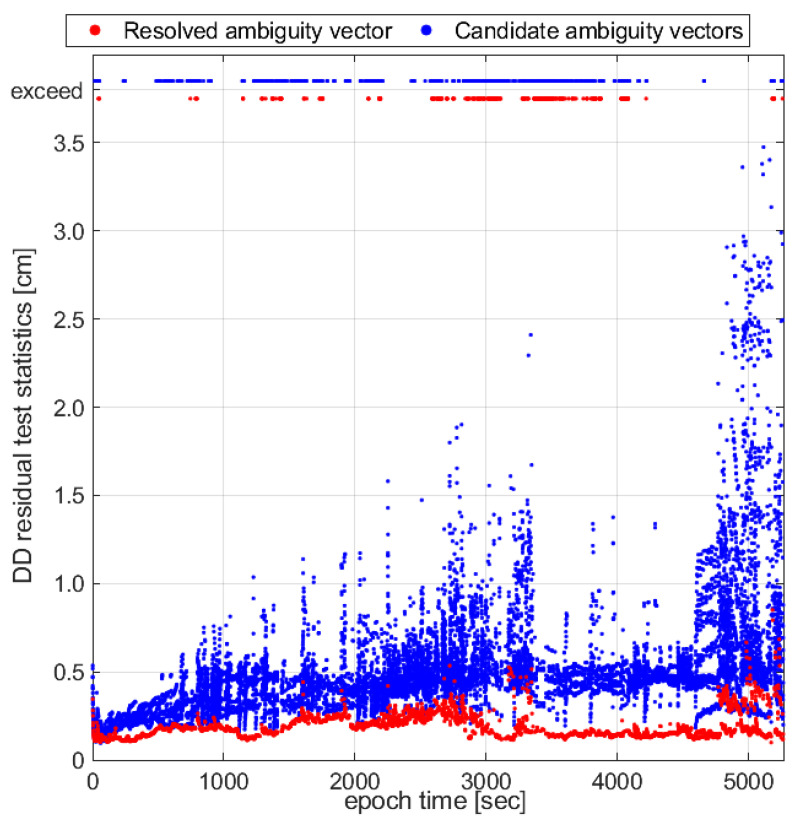
DD residual test statistics of the resolved ambiguity vector compared with candidate ambiguity vectors from the LAMBDA method in the majority test.

**Figure 5 sensors-23-05292-f005:**
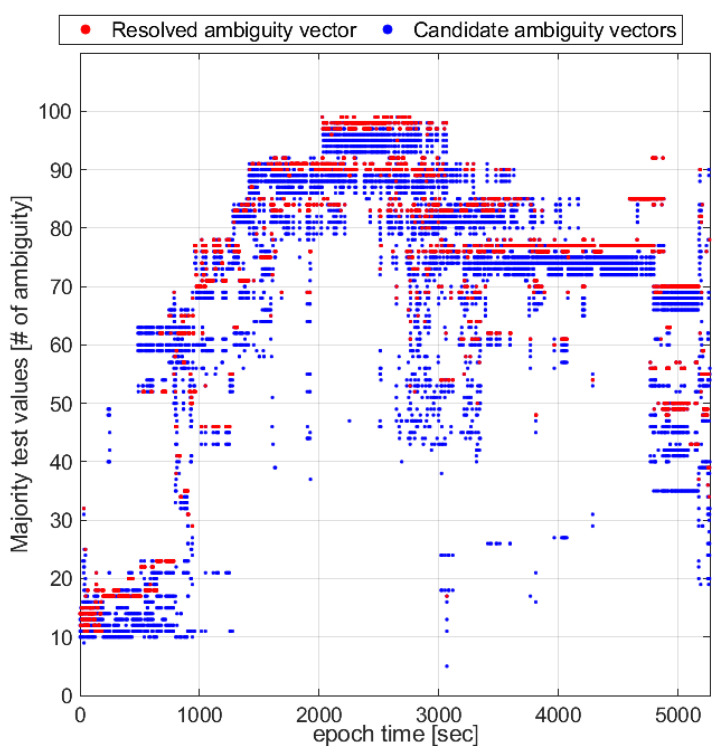
Majority test values of the resolved ambiguity vectors compared with candidate ambiguity vectors from the LAMBDA method in the majority test.

**Figure 6 sensors-23-05292-f006:**
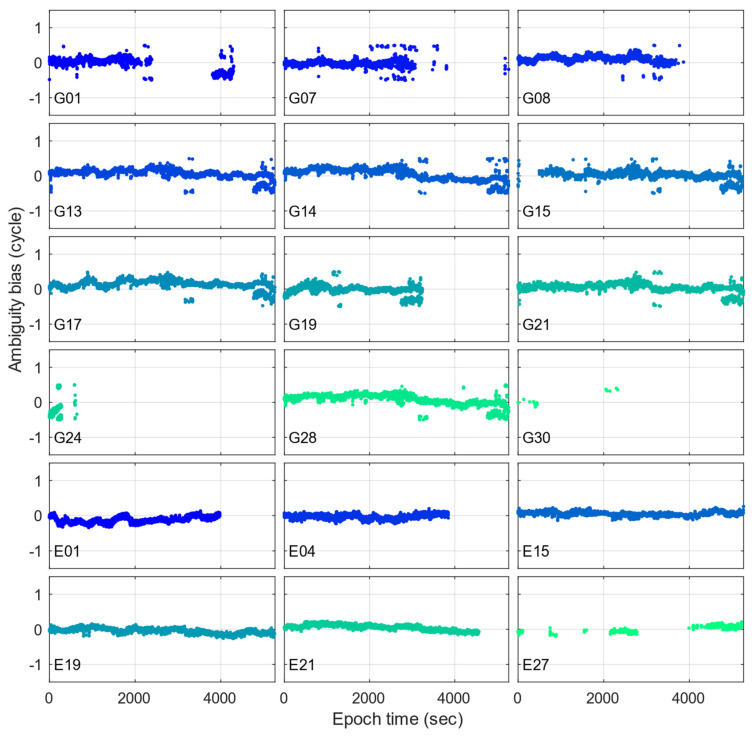
Ambiguity bias time-series for each GPS and Galileo satellite.

**Figure 7 sensors-23-05292-f007:**
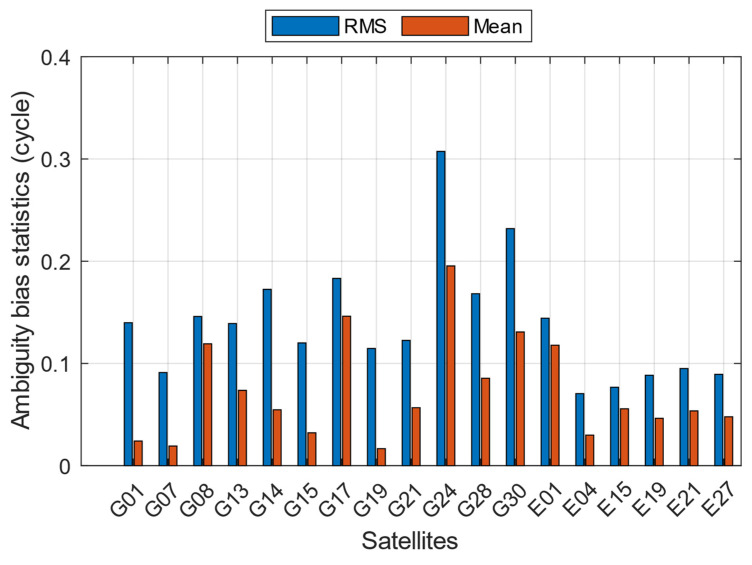
Ambiguity bias statistics for each GPS and Galileo satellite.

**Figure 8 sensors-23-05292-f008:**
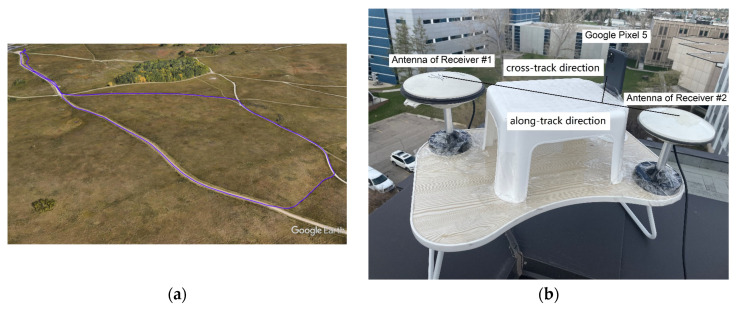
Ground trajectory (**a**) and data collection platform (**b**) of the kinematic experiment using Google Pixel 5.

**Figure 9 sensors-23-05292-f009:**
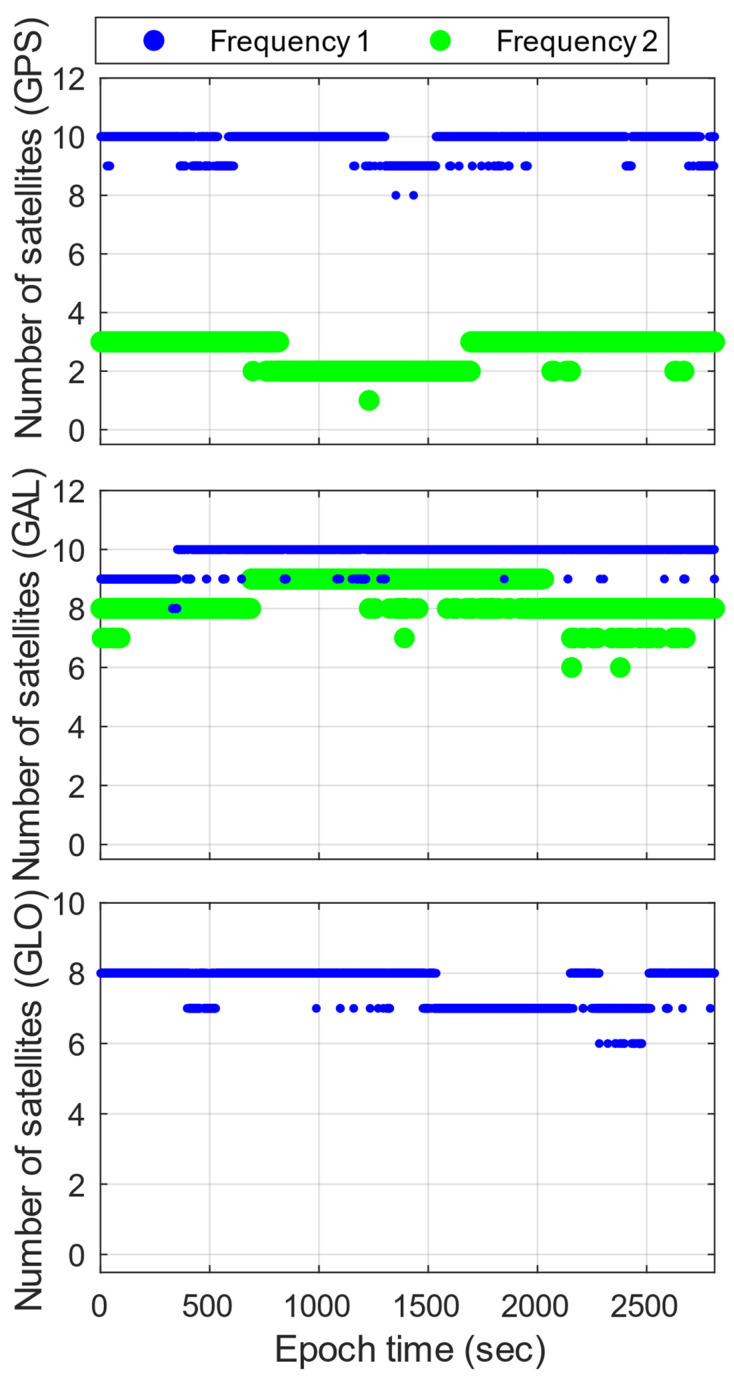
Numbers of GPS, Galileo (GAL), and GLONASS (GLO) satellites of Google Pixel 5 for dual frequency. For GPS, Frequency 1 is L1 C/A, and Frequency 2 is L5 (Q). For Galileo, Frequency 1 is E1 (C), and Frequency 2 is E5a (Q). For GLONASS, Frequency 1 is L1.

**Figure 10 sensors-23-05292-f010:**
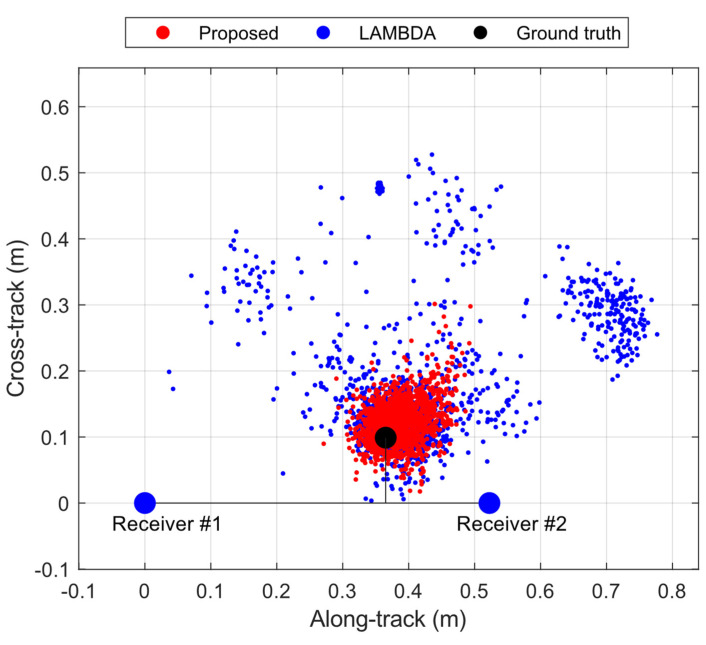
Along-track and cross-track positioning errors of Google Pixel 5 with respect to the ground truth (black dot) and antennas of two GNSS receivers (blue dots).

**Figure 11 sensors-23-05292-f011:**
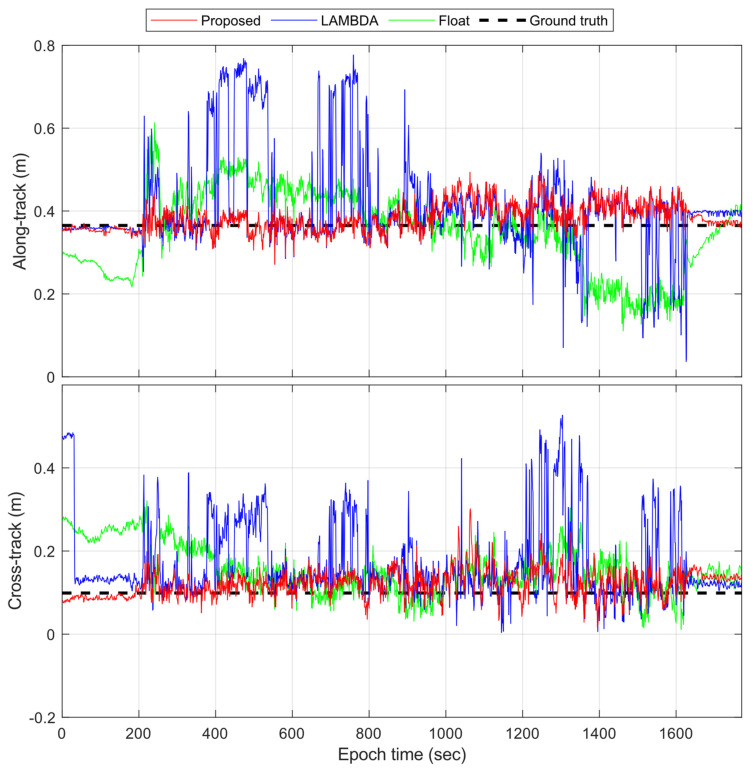
Along-track and cross-track positioning error time series of Google Pixel 5.

**Table 1 sensors-23-05292-t001:** Detailed algorithm implementation parameters for the proposed AR algorithm.

Parameter Descriptions	Values
Number of candidate vectors by LAMBDA	10 candidates
Majority test selection for candidate ambiguity vectors	First 4 candidates
Residual test selection for candidate ambiguity vectors	First 4 candidates
Residual test window size	10 epochs

**Table 2 sensors-23-05292-t002:** Filtering parameters for static and kinematic experiments.

Filtering Parameters	Values
Stochastic modelling method	Elevation-dependent model [54], $\sigma_{P}^{2}=0.5+\frac{1.5}{\sin^{2}\left(E\right)}\;m^{2}$, $\sigma_{\Phi }^{2}=10^{-6}+\frac{{4\;\times \;10}^{-6}}{\sin^{2}\left(E\right)}\;m^{2}$
Initial state variance	Coordinates $\sigma_{0,crd}^{2}=1000.0\;m^{2}$, ambiguities $\sigma_{0,amb}^{2}=1000.0\;m^{2}$
Process noise	Coordinates $Q_{crd}=20.0\;m^{2}$, ambiguities $Q_{amb}=0.00001\;m^{2}$

## Data Availability

The GNSS dataset associated with this study is publicly accessible in the GitHub repository (https://github.com/Yuting1117/Smartphone-AR-Dataset, accessed on 15 May 2023).

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
