# Peer review of "An Improved Ambiguity Resolution Algorithm for Smartphone RTK Positioning"

_sensors, 2023, doi:10.3390/s23115292_

Round 1

Reviewer 1 Report

One of the most important factors that directly affect the performance of smartphone-based positioning systems is the type of antenna used. Looking at the literature, it is seen that geodetic antennas are connected with different hardware for cm-level accuracy. In cases where its own antennas are used, sub-meter to meters accuracy is mostly obtained. In this study, sufficient information was not shared on this subject. Information should be given about this subject, namely, with which antenna the measurement is made in both static and kinematic measurements, if geodetic antenna is used, how it is connected, etc.

On the other hand, further detailing how to obtain know coordinates (for static test) and ground trajectory (for kinematic test) used to verify the performance of the proposed algorithm; It would be appropriate to explain how the processes are done (with the parameters used, etc.).

Author Response

Dear Reviewer,

We appreciate your time in reviewing our manuscript. All your corrections and comments are carefully considered and applied, please find the revised manuscript for details. Thank you again for your kind comments.

Best regards,

Yang Jiang

Comment 1: One of the most important factors that directly affect the performance of smartphone-based positioning systems is the type of antenna used. Looking at the literature, it is seen that geodetic antennas are connected with different hardware for cm-level accuracy. In cases where its own antennas are used, sub-meter to meters accuracy is mostly obtained. In this study, sufficient information was not shared on this subject. Information should be given about this subject, namely, with which antenna the measurement is made in both static and kinematic measurements, if geodetic antenna is used, how it is connected, etc.

Response 1: Thank you very much for your comment. We strongly agree that the antenna is crucial for smartphone positioning, especially in achieving AR. In our study, we aim to solve the smartphone AR based on the smartphone itself, so there is no external antenna installation in both of our experiments. We didn’t use a signal repeater to enhance the performance of smartphone observations, either. You will find that in our revision, we added this information when introducing the experiments. Please see the lines from 217 to 220 and from 331 to 332.

Comment 2: On the other hand, further detailing how to obtain know coordinates (for the static test) and ground trajectory (for the kinematic test) used to verify the performance of the proposed algorithm; It would be appropriate to explain how the processes are done (with the parameters used, etc.).

Response 2: Thank you so much for your comment. To evaluate the smartphone positioning performance, we use two survey-grade GNSS antennas coupled with ublox ZED-F9P receivers, which give RTK fixed solutions with centimeter-level accuracy. Since we have two receivers, the accuracy evaluation can only happen in two-dimensional space, which is along-track and cross-track directions. We have added some content to ensure readers can catch this information, please find the lines from 334 to 342.

Reviewer 2 Report

This manuscript presents a novel approach to ambiguity resolution for smartphones in kinematic mode.

The manuscript is very well structured and written. It is another excellent contribution from the group of articles that make an original contribution to the existing knowledge on ambiguity resolution. However, the manuscript contains some typos, namely: the use of comma instead of period (after equation 3, equation 4, equation 6, equation 9, line 178, equation 15) and a missing period (equation 14). Figures 4 and 5 may be too large. Another problem is in the citation: in line 305 it would be advisable to give a reference, not just the year (consecutive number in square brackets), and the same is true for lines 378 and 379. The caption of Figure 8 should be changed by indicating what the (a) and (b) figure is.

Be that as it may, the proposed algorithm and results will certainly have an impact on the processing of GNSS data for signals from smartphones in the future, but more research should be done with more datasets from smartphones. In my opinion, the manuscript is well suited to be accepted for publication as is.

Finally, I would like to congratulate the authors for their work.

Author Response

Dear Reviewer,

We appreciate your time in reviewing our manuscript. All your corrections are carefully considered and applied, please find the revised manuscript for details. Thank you again for your kind comments, we look forward to any further questions you have.

Best regards,

Yang Jiang

Reviewer 3 Report

Corrections are given in the PDF document.

Moderate editing of English language required.

Author Response

Dear Reviewer,

We appreciate your time for reviewing our manuscript. All your corrections are carefully considered and applied (please find the revised manuscript for details), except a couple of them:

  • Lines 157-159: add: [Reference]

The modulus of each ambiguity ai, namely  Mai, is calculated from the combined candidate vector A by the LAMBDA method, formulated as follows [Reference]:

  • Lines 168-169: add: [Reference]

To exclude the abnormal candidate ambiguities, the majority test is also applied to each candidate ambiguity vector, formulated as follows [Reference]

  • Lines 168-169: add: [Reference]

To exclude the abnormal candidate ambiguities, the majority test is also applied to each candidate ambiguity vector, formulated as follows [Reference]:

For these two places, there are no references because these are the original proposal in this study. To ensure the readers’ awareness, we have made some revisions to the text, please see the revised manuscript for details:

  • Lines 157-159,
  • Lines 168-170,
  • Lines 184-185.

Thanks again for your kind comments, please let us know if you have any questions.

Best regards,

Yang Jiang